# Boosting Learning for LDPC Codes
# to Improve the Error-Floor Performance

**Hee-Youl Kwak**
University of Ulsan
ghy1228@gmail.com

**Dae-Young Yun**
Seoul National University
bigbowl204@snu.ac.kr

**Yongjune Kim**[*]
POSTECH
yongjune@postech.ac.kr

**Sang-Hyo Kim**[*]
Sungkyunkwan University
iamshkim@skku.edu

**Jong-Seon No**
Seoul National University
jsno@snu.ac.kr

## Abstract

Low-density parity-check (LDPC) codes have been successfully commercialized in communication systems due to their strong error correction capabilities and simple decoding process. However, the error-floor phenomenon of LDPC codes, in which the error rate stops decreasing rapidly at a certain level, presents challenges for achieving extremely low error rates and deploying LDPC codes in scenarios demanding ultra-high reliability. In this work, we propose training methods for neural min-sum (NMS) decoders to eliminate the error-floor effect. First, by leveraging *the boosting learning technique* of ensemble networks, we divide the decoding network into two neural decoders and train the post decoder to be specialized for uncorrected words that the first decoder fails to correct. Secondly, to address the vanishing gradient issue in training, we introduce a *block-wise training schedule* that locally trains a block of weights while retraining the preceding block. Lastly, we show that assigning different weights to unsatisfied check nodes effectively lowers the error-floor with a minimal number of weights. By applying these training methods to standard LDPC codes, we achieve the best error-floor performance compared to other decoding methods. The proposed NMS decoder, optimized solely through novel training methods without additional modules, can be integrated into existing LDPC decoders without incurring extra hardware costs. The source code is available at `https://github.com/ghy1228/LDPC_Error_Floor`.

## 1  Introduction

The field of learning-based decoding for error-correcting codes began with research on training neural networks to produce the information vector when given a distorted codeword [1, 2, 3, 4]. These works assume an arbitrary neural network with no prior knowledge of decoding algorithms, and accordingly, face the challenge of learning a decoding algorithm. In contrast, model-based neural decoders are designed by mapping a well-known graph-based iterative decoding algorithm, such as belief propagation (BP) and min-sum (MS) decoding algorithms, to a neural network and then training its weights [5]. Compared to the arbitrary network approaches or the error correction transformer [6], model-based neural decoders offer the advantages of guaranteeing the performance of existing iterative algorithms and using hardware architectures [7] that are already well optimized for iterative decoding algorithms.

---

[*]Corresponding authors.

37th Conference on Neural Information Processing Systems (NeurIPS 2023).

LDPC codes have been incorporated into WiMAX and 5G communication systems [8, 9], owing to their strong error-correcting capabilities and low decoding complexity [10, 11]. However, more advanced LDPC coding technology needs to be developed for diverse communication environments lying in the scope of future 6G systems. In particular, for environments that require extremely low frame error rate (FER) such as the next generation ultra-reliable and low-latency communications (xURLLC) [12], it is crucial to mitigate the error-floor in the decoding of LDPC codes. The error-floor phenomenon refers to an abnormal phenomenon where the FER does not decrease as rapidly as in the waterfall region [11, 13]. The error-floor phenomenon also should be addressed for systems demanding very high reliability, such as solid-state drive (SSD) storage [14], DNA storage [15], and cryptosystems [16]. However, enhancing other features of LDPC codes often inadvertently reinforces the error-floor phenomenon as a side effect. For instance, the error-floor tends to be intensified when optimizing LDPC codes for superior waterfall performance or decoding by low complexity decoders such as quantized MS decoders [17]. Therefore, research focused on alleviating the error-floor, especially when decoding LDPC codes optimized for performance with low decoding complexity, has become significant. Such advancements will broaden the applications of LDPC codes.

## 1.1 Main contributions

With this need in mind, we focus on how to train a low-complexity neural MS (NMS) decoder to prevent the error-floor in well designed LDPC codes. The main contributions of the paper are threefold as follows.

*Boosting learning using uncorrected words:* We first leverage the boosting learning technique [18, 19] that employs a sequential training approach for multiple classifiers, wherein subsequent classifiers concentrate on the data samples that preceding classifiers incorrectly classify. Inspired by this method, we divide the neural decoder into two cascaded neural decoders and train the first decoder to be focused on the waterfall performance, while training the second decoder to be specialized in handling the uncorrected words that are not corrected by the first decoder due to the error-floor phenomenon. Uncorrected words in the error-floor region mostly contain small-error patterns related to trapping sets or absorbing sets [11], which can be effectively corrected by weighting decoding messages. As a result, a significant performance improvement in the error-floor region is achieved by boosting learning.

*Block-wise training schedule with retraining:* To mitigate the error-floor, iterative decoding typically requires a large number of decoding iterations, often exceeding 50 [17, 20, 21, 22]. However, NMS decoders encompassing many iterations can undergo the vanishing gradient problem in training [23]. To address this problem, we propose a new training schedule inspired by block-wise training methods [24, 25]. The proposed block-wise training schedule divides the entire decoding iterations into sub-blocks and trains these sub-blocks in a sequential manner. Additionally, rather than fixing the weights trained from previous blocks, we retrain them to escape from local minima. As a result, the proposed schedule enables to train numerous weights for all 50 iterations successfully while outperforming both the multi-loss method [5] and the iter-by-iter schedule [26].

*Weight sharing technique with dynamic weight allocation:* The weight sharing technique is a way to reduce the number of trainable weights by grouping specific weights to share a common value. The waterfall performance does not severely degrade even if we bundle all weights for each iteration [27, 26]. However, our observations indicate that this does not hold true in the error-floor region, implying that a higher degree of weight diversity is necessary to correct error patterns causing the error-floor. To obtain sufficient diversity with a minimal number of weights, we dynamically assign different weights to unsatisfied check nodes (UCNs) and satisfied check nodes (SCNs) in the decoding process. By utilizing only two weight values for SCNs and UCNs each iteration, we achieve the performance of the NMS decoder using different weights for every edge. This method reduces the number of weights to be trained to only 2.6% of the original number of weights.

We employ these training methods on a range of representative LDPC codes adopted for standards such as WiMAX [8], IEEE 802.11n [28], and 5G new radio [9]. The FER point at the onset of the error-floor diminishes by over two orders of magnitude for all codes compared to conventional weighted MS (WMS) decoding [29]. Compared to existing NMS decoding approaches [27, 26], our proposed scheme exhibits a notably enhanced capability to suppress the error-floor. This scheme also achieves a similar performance as the state-of-the-art post-processing method in [22], with only a third of the iterations.

Table 1: Comparison between model-based neural decoders

| Reference | Codes | Target region | Decoders | Training sample | Training schedule | Weight sharing |
|---|---|---|---|---|---|---|
| This work | Standard LDPC | Waterfall, Error-floor | MS | Uncorrected words | Block-wise with retraining | Spatial with UCN weights |
| [5] | BCH | Waterfall | BP, MS | Received words | One-shot | Temporal |
| [27] | Standard LDPC | Waterfall | BP, MS | Received words | Iter-by-Iter | Spatial |
| [30] | Short LDPC | Waterfall | BP | Absorbing set | One-shot | Temporal |
| [31] | Regular LDPC | Waterfall, Error-floor | FAID | Received words | One-shot | Temporal |
| [32] | Short LDPC | Waterfall, Error-floor | FAID | Trapping set | One-shot | Temporal |
| [33] | Standard LDPC | Waterfall | Layered | Received words | Iter-by-Iter | UCN weights |

## 1.2 Related works

We compare the proposed training scheme with the existing schemes for NMS decoders in Table 1. First, all works except [31, 32] aim to improve the waterfall performance. Although the scope of the works in [31, 32] includes the error-floor performance, they assumed specific conditions of binary symmetric channels (BSC) and FAID, while we deal with the more general situation of additive white Gaussian noise (AWGN) channels and MS decoding. Regarding the training sample selection, the training samples can be received words randomly taken from the AWGN channel [5, 27, 31, 33], or codewords with erroneous trapping sets (or absorbing sets) [30, 32]. However, to use the method in [30, 32], trapping sets should be enumerated, which is only applicable to short LDPC codes and not feasible for medium to large length standard LDPC codes. In contrast, the proposed boosting method, which generates training samples through decoding with linear complexity, can be applied even to codes of several thousand lengths.

For scheduling of training, it is common to train all weights at once (One-shot training) [5] and some works sequentially train the weights corresponding to a single iteration locally [27, 33], while we train a block of iterations with retraining. In terms of the weight sharing techniques, we confirm that the proposed sharing technique using UCN weights is superior to the spatial or temporal sharing technique used in [5, 27, 30, 31, 32]. Meanwhile, a method of assigning different weights to UCNs has been introduced in [33], but they applied the same UCN weight to all CNs belonging to a proto CN when at least one CN is unsatisfied, whereas we pinpoint specific UCNs and apply weights individually.

There have been studies adding hypernetworks to the Vanilla NMS decoder [34, 35, 36] or using a transformer architecture [6] to improve the waterfall performance at the expense of increased training and decoding costs. While the proposed training techniques are broadly applicable to these augmented neural decoders, this work primarily aims to improve the error-floor performance of the Vanilla NMS decoder under practical conditions.

## 2 Preliminaries

### 2.1 LDPC codes

In this paper, we consider quasi-cyclic (QC) LDPC codes, which have been adopted in various applications due to their implementation advantages [13, 37]. The Tanner graph of a QC-LDPC code, consisting of $n = Nz$ VNs and $m = Mz$ CNs, can be obtained by lifting the protograph, composed of $M$ proto CNs and $N$ proto VNs, with a lifting factor $z$ [37, 38]. Let $E$ be the total number of edges in the protograph. As a running example, we use the WiMAX QC-LDPC code of length $n = 576$ and code-rate $3/4$ with $N = 24, M = 6, E = 88$, and $z = 24$ [8] .

### 2.2 Neural min-sum decoding

For iteration $\ell$, let $m_{c \to v}^{(\ell)}$ represent the message from CN $c$ to VN $v$ and let $m_{v \to c}^{(\ell)}$ represent the message from VN $v$ to CN $c$. The neighboring nodes of $x$ are represented by $\mathcal{N}(x)$. The initial conditions are $m_{v \to c}^{(0)} = m_v^{\text{ch}}, m_{c \to v}^{(0)} = 0$ for the channel LLR $m_v^{\text{ch}}$ of VN $v$. For $\ell = 1, \ldots, \bar{\ell}$, the

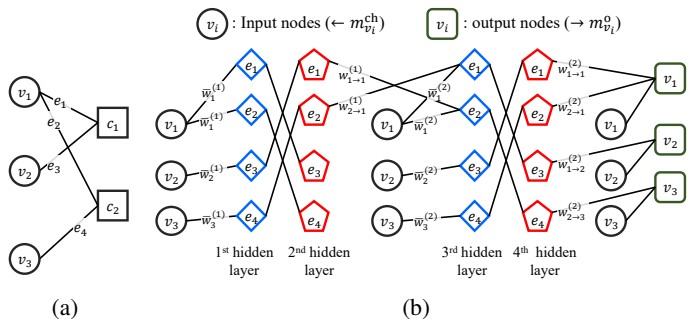

Figure 1: (a) The Tanner graph of an LDPC code and (b) the neural network corresponding to NMS decoding with a maximum iteration of $\bar{\ell} = 2$.

NMS decoding algorithm [5] updates the messages as follows

$$m_{v \to c}^{(\ell)} = \overline{w}_v^{(\ell)} m_v^{\text{ch}} + \sum_{c' \in \mathcal{N}(v) \backslash c} m_{c' \to v}^{(\ell-1)} \tag{1}$$

$$m_{c \to v}^{(\ell)} = w_{c \to v}^{(\ell)} \left( \prod_{v' \in \mathcal{N}(c) \backslash v} \text{sgn}\left( m_{v' \to c}^{(\ell)} \right) \right) \min_{v' \in \mathcal{N}(c) \backslash v} |m_{v' \to c}^{(\ell)}|, \tag{2}$$

where $\overline{w}_v^{(\ell)}$ and $w_{c \to v}^{(\ell)}$ are called the VN weight and CN weight, respectively. At the last iteration $\bar{\ell}$, output LLRs $m_v^{\text{o}}$ are computed as $m_v^{\text{o}} = m_v^{\text{ch}} + \sum_{c' \in \mathcal{N}(v)} m_{c' \to v}^{(\ell)}$.

By quantizing $m_{v \to c}^{(\ell)}$, $m_{c \to v}^{(\ell)}$, and $m_v^{\text{ch}}$, the quantized NMS decoding algorithm is obtained. The quantized decoders are widely used in practical applications due to its low complexity and commonly employed in existing error-floor researches [17, 20, 21, 22]. Therefore, we use it to ensure a fair comparison. Specifically, we use 5-bit uniform quantization with a maximum magnitude of 7.5 and a step size of 0.5 for the quantized NMS decoder as in [20, 21, 22].

### 2.3 Training weights for the NMS decoder

If all the weights in (1) and (2) are set to 1, NMS decoding is equivalent to MS decoding [39], or if VN weights $\overline{w}_v^{(\ell)}$ are 1 and CN weights $w_{c \to v}^{(\ell)}$ have the same value, the decoder operates as the WMS decoder [40]. The NMS decoder gains performance improvement over the WMS or MS decoder by greatly increasing the diversity of weights. However, the full diversity weights increase the training complexity and require a large amount of memory to store the weights. Therefore, previous studies used weight sharing techniques by assigning the same value to weights with the same attributes. First, since this paper deals with QC-LDPC codes, we use the protograph weight sharing technique [26] by default, assigning the same weight value to the VNs (or CNs) belonging to a proto VN (or CN). Then, the weights to be trained are represented by $\{\overline{w}_{v_p}^{(\ell)}, w_{c_p \to v_p}^{(\ell)}\}$ for a proto VN $v_p$ and a proto CN $c_p$. The total number of weights is then $(N + E)\bar{\ell}$. If we employ spatial weight sharing in [27], only one VN weight and one CN weight remain for each iteration, and the weights are $\{\overline{w}^{(\ell)}, w^{(\ell)}\}_{\ell=1}^{\bar{\ell}}$, with a total number of $2\bar{\ell}$. On the other hand, by using temporal weight sharing [5] to eliminate differences between iterations, the weights are $\{\overline{w}_{v_p}, w_{c_p \to v_p}\}$, and the number is $(N + E)$.

The neural network in Fig. 1(b) corresponds to NMS decoding of the Tanner graph in Fig. 1(a) with $\bar{\ell} = 2$. The input to this neural network is the channel LLR vector $(m_1^{\text{ch}}, \ldots, m_n^{\text{ch}})$, and the output is the output LLR vector $(m_1^{\text{o}}, \ldots, m_n^{\text{o}})$. For each iteration, two hidden layers are arranged, and each hidden layer has a number of nodes equal to the number of edges in the Tanner graph. In the odd hidden layers, the VN to CN message operation in (1) is performed, while in the even hidden layers, the CN to VN message operation in (2) is performed. The input layer is also connected to the odd hidden layers, which corresponds to the addition of the channel LLR in (1). The messages from the $2\ell$-th hidden layer to the $(2\ell + 1)$-th hidden layer are weighted by $w_{c \to v}^{(\ell)}$, and the messages from the

input nodes to the $(2\ell + 1)$-th hidden layer are weighted by $\overline{w}_v^{(\ell)}$. As our goal is to reduce the FER in the error-floor region, we use the FER loss, $\frac{1}{2}\left[1 - \text{sgn}\left(\min_{1 \le v \le N} m_v^o\right)\right]$ [32].

# 3 Proposed training method

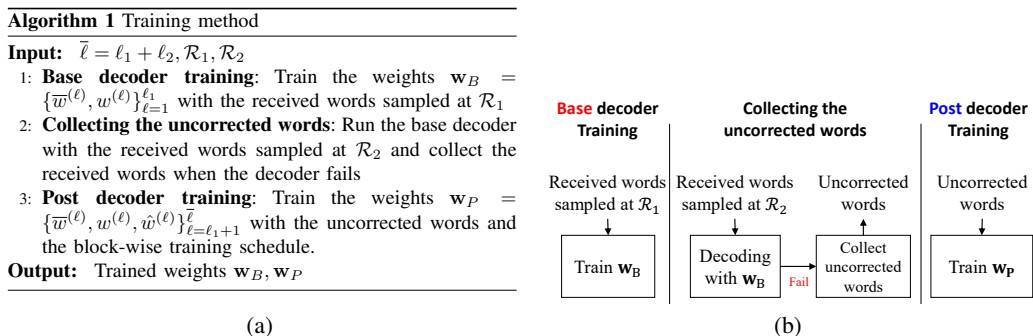

(a)                (b)

Figure 2: The proposed training method represented by (a) an algorithm and (b) a block diagram.

In this section, we introduce the proposed training methods through three subsections. The organized training algorithm is shown in Fig. 2.

## 3.1 Boosting learning using uncorrected words

For the boosting learning approach, we first divide the entire decoding process into two stages: the base decoding stage $\{1, \ldots, \ell_1\}$ and the post decoding stage $\{\ell_1 + 1, \ldots, \ell_1 + \ell_2 = \overline{\ell}\}$. Training of base decoder follows the conventional training method: the received words sampled from $E_b/N_0$ region $\mathcal{R}_1$ are used as training samples. Specifically, we set $\ell_1 = 20$ and $\mathcal{R}_1 = \{2.0, 2.5, 3.0, 3.5, 4.0\}$, which corresponds the waterfall region of WMS decoding. We use the spatial sharing technique (i.e., $\mathbf{w}_B = \{\overline{w}^{(\ell)}, w^{(\ell)}\}_{\ell=1}^{\ell_1}$) since this achieves comparable performance to the full diversity weights in the waterfall region.

In Fig. 3(a), the base NMS decoder is compared with the MS decoder and the WMS decoder with a single weight of $0.75$ for $\overline{\ell} = 20$. The WMS decoding performance has a severe error-floor even though its waterfall performance is better than the MS decoding performance. Compared to the MS and WMS decoders, the NMS decoder for $\overline{\ell} = 20$ performs better over the training range $\mathcal{R}_1$. On the other hand, the NMS decoder performs worse than the MS decoder in the error-floor region (e.g., $4.5$ dB), which is outside $\mathcal{R}_1$. To improve the performance in the error-floor region, a straightforward approach is extending the training range $\mathcal{R}_1$ to include the error-floor region. However, the FER of the base decoder for received words from the error-floor region is very low, resulting in an almost negligible FER loss. Consequently, integrating the error-floor region into the training range does not impact the weight update process.

Before training the post decoder, we first collect uncorrected words that the trained base decoder fails to correct among the received words sampled from region $\mathcal{R}_2$. Then, the uncorrected words serve as training samples for the post decoder, which is distinct from the conventional training methods. The post decoder trains the weights $\{\overline{w}_{v_p}^{(\ell)}, w_{c_p \to v_p}^{(\ell)}\}_{\ell=\ell_1+1}^{\overline{\ell}}$ with the aim of correcting the uncorrected words. After completing the training, the trained weights are used for the NMS decoding algorithm in (1)–(2). From the perspective of the NMS decoder, it performs continuous decoding up to iteration $\overline{\ell}$ using the trained weights, but for the sake of discussion, we assume as if there are two cascaded decoders (base and post) in the perspective of training. Note that we employ the full diversity weights for the post decoder to confirm the best performance but we will introduce the shared weights $\mathbf{w}_P$ (used in Fig. 2) in the next subsection. We also set $l_2 = 10$, $\overline{\ell} = 30$ for this experiment, and subsequently extend the maximum number of iterations in the following subsection.

To analyze the effectiveness of the proposed boosting learning, we compare the following three cases.

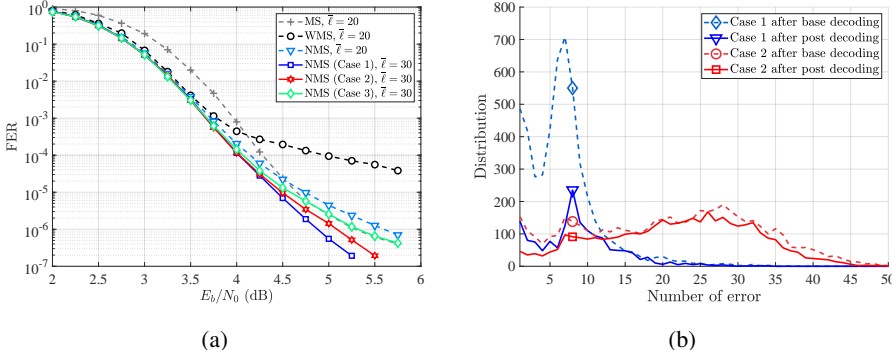

(a)                                                                  (b)

Figure 3: (a) Decoding performances of the MS, WMS, NMS decoders and (b) Error distributions after base and post decoding for Case 1 and Case 2.

Case 1  Uncorrected words sampled at $4.5$ dB in the error-floor region (i.e., $\mathcal{R}_2 = 4.5$).

Case 2  Uncorrected words sampled at $3.5$ dB in the waterfall region (i.e., $\mathcal{R}_2 = 3.5$).

Case 3  Received words sampled at $4.5$ dB without filtering.

Regarding Case 1 and Case 2, we collect a total of $60{,}000$ uncorrected words, allocating $50{,}000$ for training, $5{,}000$ for validation, and remaining $5{,}000$ for test. Training is conducted for $100$ epochs. Fig. 3(b) shows the distribution of the number of errors after base decoding and post decoding for the test samples used in Case 1 and Case 2. For Case 1, the uncorrected words collected in the error-floor region mainly have a small number of errors since most of decoding errors are trapped in small trapping sets, so the distribution is concentrated on small numbers (see Case 1 after base decoding). For ease of use, we refer to codewords with fewer than $11$ remaining errors as small-error words. The post decoder, which is mainly trained on these small-error words, corrects a significant number of small-error words (see Case 1 after post decoding). Out of the total $5{,}000$ test samples, $68.5\%$ of samples are corrected by the post decoder, resulting that the test FER for the test samples is $0.315$. This means that, when decoding for received words of the AWGN channel, the resulting FER at $E_b/N_0 = 4.5$ dB after post decoding is $0.315$ times of the FER after base decoding as shown in Fig 3(a). In other words, the post decoder is successfully trained to correct small-error words inducing the error-floor.

On the other hand, Case 2, where the samples are collected from the waterfall region, has a distribution that is widespread across all areas (see Case 2 after base decoding). In addition, the distribution remains almost the same after post decoding (see Case 2 after post decoding), which means that the post decoder fails to reduce the FER sufficiently. For the test samples, the test FER at $E_b/N_0 = 3.5$ dB after post decoding is $0.77$ times of the FER after base decoding whose difference is not noticeable as shown in Fig 3(a). Comparing the results of the two cases, we conclude that composing mainly with small-error words facilitates the post decoder to learn to correct small-error words more effectively. As a result, Fig. 3(a) shows that Case 1 mitigates the error-floor more than Case 2. Meanwhile, for Case 3, where all received words are used as training samples without filtering, almost all of them are corrected during base decoding. Since the post stage training is mainly performed on codewords without errors, the loss function becomes almost $0$. Then, the weights of the post decoder are unchanged from the initial value $1$, and accordingly, the performance approaches the MS decoding performance, as shown in Fig. 3(a).

## 3.2  Block-wise training schedule

In the previous subsection, the number of iterations for the post decoder is set to $\ell_2 = 10$. To lower the error-floor further, a large number of iterations is required, so we set $\ell_2 = 30, \bar{\ell} = 50$. However, deep neural decoders with a large iteration number are prone to suffer from the vanishing gradient problem. In order to tackle this issue, we propose a block-wise training schedule which is shown in Fig. 4(a). The proposed training schedule locally trains the weights corresponding to a block of $\Delta_1$ iterations at each training stage. In the first stage, the weights belonging to the first block are trained,

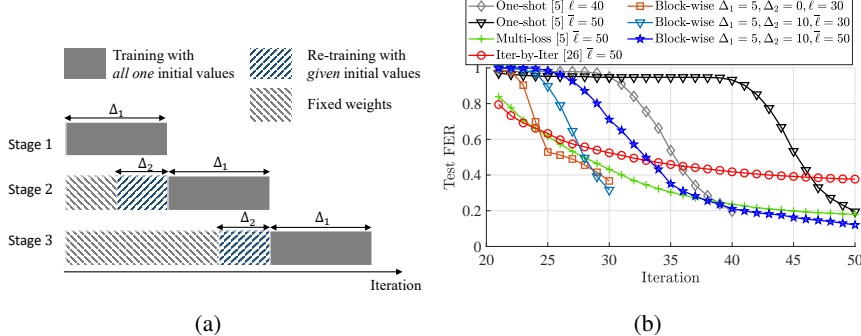

Figure 4: (a): Block-wise training schedule and (b): Evolution of the test FER across iterations.

| | Test FER | Number of weights |
|---|---|---|
| Full diversity | 0.112 | $(N + E)\ell_2 = 3360$ |
| Spatial sharing | 0.168 | $2\ell_2 = 60$ |
| Temporal sharing | 0.186 | $(N + E) = 112$ |
| Proposed sharing | 0.111 | $3\ell_2 = 90$ |

Figure 5: Illustration of the proposed weight sharing technique and comparison with other sharing techniques.

and in the next stage, the weights of the subsequent $\Delta_1$ iterations are trained. At this point, the weight values of previous $\Delta_2$ iterations, which are already trained in the first stage, are further trained by taking the result of the first stage training as its initial state. This retraining, which is not used in the iter-by-iter training schedule [26], assists in preventing the learning process from falling into a local minimum. Note that the method of one-shot training [5] corresponds to the case of $\Delta_1 = \ell_2, \Delta_2 = 0$, and the iter-by-iter training schedule [26] is equivalent to the case of $\Delta_1 = 1, \Delta_2 = 0$.

We employ a greedy approach to determine the optimal values for $\Delta_1$ and $\Delta_2$, resulting that $\Delta_1 = 5, \Delta_2 = 10$ offers superior performance in terms of the test FER. Fig. 4(b) shows the evolution of test FER as a function of the iteration number. In the case of one-shot training [5], the vanishing gradient problem hinders training the weights of earlier iterations, so the test FER stays roughly the same until iteration 40 and starts to fall thereafter. The same behavior is observed for $\bar{\ell} = 40$. Thus, the test FERs of $\bar{\ell} = 40$ and $\bar{\ell} = 50$ are almost the same. Next, since the iter-by-iter training schedule [26] finds the optimal weight for each individual iteration, the test FER falls even at the first iteration of the post decoder (i.e., $\ell = 21$) without encountering the vanishing gradient problem. However, this local optimization leads to a degraded local minimum, and consequently, the test FER gradually and slowly decreases with each iteration. Likewise, the multi-loss method [5] shows a similar result. In contrast, the block-wise training schedule with $\Delta_1 = 5, \Delta_2 = 0$ shows a superior test FER at iteration 25 compared to the other training schedules because it results in a better solution by training the weights of multiple iterations simultaneously. Moreover, the schedule with retraining (i.e., $\Delta_1 = 5, \Delta_2 = 10$) outperforms the schedule without retraining (i.e., $\Delta_1 = 5, \Delta_2 = 0$) at iteration 30 though it shows a worse result at iteration 25. This implies that through retraining, the weights of intermediate iterations have been adjusted to preprocess error patterns thereby leading to stronger correction capabilities in the final iteration. As a result, at the maximum of 50 iterations, the proposed training schedule with $\Delta_1 = 5, \Delta_2 = 10$ provides the better test FER value compared to the other training schedules as shown in Fig. 4(b): 0.11 for the block-wise, 0.16 for the multi-loss, 0.18 for the one-shot, 0.37 for the iter-by-iter, .

### 3.3 Weight sharing technique using UCN weights

Assuming the techniques proposed thus far (in detail, uncorrected words at $E_b/N_0 = 4.5$ dB and training schedule with $\Delta_1 = 5, \Delta_2 = 10$) are used, we compare the weight sharing techniques

in Fig. 5. Compared to the full diversity weights, the spatial and temporal sharing techniques significantly reduce the number of distinct weights, but cause performance degradation. In contrast, the proposed sharing technique that introduces a new weight type called UCN weight shows almost identical performance while using only about 2.6% of the weights compared to the full diversity weights. The proposed sharing technique assigns different weights to UCNs and SCNs as shown in Fig. 5. This is feasible because the decoder knows whether a CN satisfies the check equation or not. Using the spatial sharing technique and distinguishing between SCN weight $w^{(\ell)}$ and UCN weight $\hat{w}^{(\ell)}$, the proposed sharing technique can be represented as $\{\overline{w}^{(\ell)}, w^{(\ell)}, \hat{w}^{(\ell)}\}_\ell$ for iteration $\ell$ and the total number of distinct weights becomes $3\ell_2$.

Techniques using different weights for UCNs and SCNs have also been proposed in [33, 41]. However, the work [41] uses only one suppression factor $\rho$ to represent the UCN weight (i.e., $\hat{w}^{(\ell)} = (1 + \rho)w^{(\ell)}$), making the UCN weight dependent on the CN weight. As a result, due to the limited degree of freedom for the UCN weight, it is difficult to obtain the decoding diversity for effectively removing various types of error patterns. Moreover, in [33], if at least one of the CNs belonging to a single proto CN is unsatisfied, all $z$ CNs from the proto CN are weighted by the UCN weight. This approach, which applies the same weight to a large number of CNs tied together at the proto level, is not suitable for correcting words with a small number of UCNs, because it does not separately handle individual CNs like the proposed method.

## 4  Performance evaluation

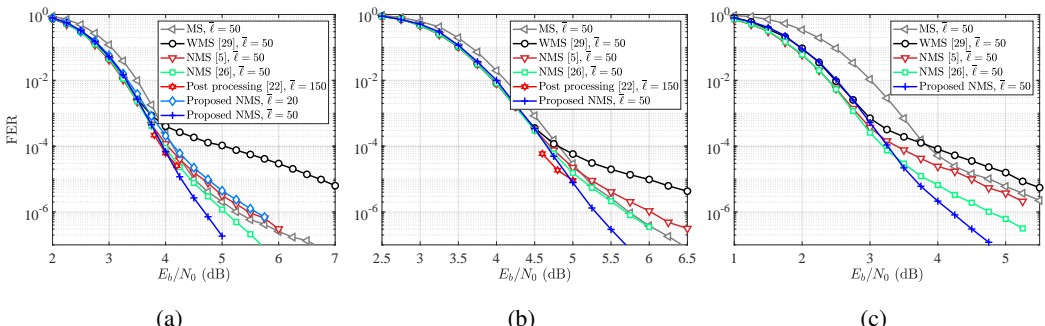

Figure 6: FER performances of (a): WiMAX LDPC (length $576$, rate $3/4$), (b): IEEE802.11n LDPC (length $648$, rate $5/6$), (c): 5G LDPC (length $552$, rate $1/2$) codes.

In this section, we compare the proposed and other conventional decoding schemes in terms of the decoding performance. All simulations are performed by NVIDIA GeForce RTX 3090 GPU and AMD Ryzen 9 5950X 16-Core Processor CPU. For training, the weights are trained by the Adam optimizer [42] with a learning rate of $0.001$. We evaluate the decoding performance using Monte Carlo methods with at least $500$ uncorrected words for each FER point. Fig. 6(a) shows the FER performances of the proposed scheme for the WiMAX LDPC code. The proposed scheme incorporates the i) boosting learning with uncorrected words, ii) block-wise training schedule with retraining, and iii) spatial weight sharing with UCN weights. The performance is compared with MS decoding, WMS decoding, and existing NMS decoding schemes in [5, 26]. Among the neural decoder studies listed in Table 1, we exclude comparison with the studies that use FAID and layered decoding, or that require enumerating trapping sets and absorbing sets. In addition, we choose not to compare with augmented neural networks [6, 34] since our approach does not increase model complexity to deal with the low-error-rate region of long codes. A comparative analysis for short codes in the waterfall region can be found in the appendix.

For the NMS decoding schemes in [5, 26], the base decoder is used for iterations from 1 to 20 like the proposed scheme, and the training methods introduced in [5, 26] are employed for the post stage. The full diversity weights are used for the schemes in [5, 26] to observe their best performance. For the scheme in [5], received words in the waterfall region ($E_b/N_0$ 2-4 dB) are used as training samples, and the weights for the post stage are trained all at once without a training schedule. For the scheme

Table 2: Complexity comparison for the WiMAX LDPC of $(N, M, E, z, \alpha) = (24, 6, 88, 24, 15.6)$, where $\alpha$ is the comparison count for each CN (i.e., $\alpha = d_c + \lceil \log d_c \rceil - 2$ for the CN degree $d_c$ [13]).

| | Complexity per iteration | | | Total complexity $(A + 2C + M)\bar{\ell}$ | Memory for weights |
|---|---|---|---|---|---|
| | Addition $A$ | Comparison $C$ | Multiplication $M$ | | |
| MS, $\bar{\ell} = 50$ | $2Ez$ $= 4224$ | $\alpha M z$ $= 2256$ | $Ez = 2112$ | 542400 | 0 |
| NMS [5, 26], $\bar{\ell} = 50$ | | | $(2E + N)z$ $= 4800$ | 676800 | 3760 |
| Proposed NMS, $\bar{\ell} = 50$ | | | | | 130 |
| Post processing [22] $\bar{\ell} = 150$ | | | $Ez = 2112$ | 1627200 | 0 |

in [26], received words from the $E_b/N_0$ points where the MS decoder achieves bit error rate of $10^{-3}$ are used as training samples, and the iter-by-iter training schedule is employed. The remaining hyper-parameters are set in the same way as in the proposed scheme. As shown in Fig. 6(a), the conventional NMS decoders in [5, 26] show good performance in the waterfall region ($E_b/N_0$ 2-4 dB), but the error-floor occurs from $4$ dB. This is because the training samples are composed of received words without filtering. In contrast, the proposed scheme shows excellent performance in both the waterfall and error-floor regions, and the error-floor phenomenon is barely noticeable down to FER of $10^{-7}$. In particular, comparing the results of $\bar{\ell} = 20$ and $\bar{\ell} = 50$, it is confirmed that the post decoder successfully removes the error-floor.

In addition, we compare the proposed scheme with the state-of-the-art post processing scheme in [22]. We directly reference the simulation results from [22]. As shown in Fig. 6(a), the scheme in [22] shows a similar or worse performance to the proposed scheme, but it has disadvantages of having very high decoding complexity and latency since it consumes a large number of iterations $\bar{\ell} = 150$. Table 2 compares the schemes in terms of the decoding complexity. The NMS decoder has more multiplications than the MS decoder by $(E + N)z$ due to the weighting operation. The number of other operations is the same as in the MS decoder. Total complexity is evaluated with assumption that the comparison $C$ is as twice as complex than the addition $A$ and multiplication $M$ [43]. The additional memory for storing the weights of the proposed scheme is $3\bar{\ell}$ which is much lower than those of [5, 26] which exploit full weight diversity. Since the scheme in [22] does not use weighting, the complexity per iteration is lower than the proposed NMS scheme, but the total complexity is more than twice as high as the proposed NMS scheme due to the higher number of iterations. Moreover, additional complexity is required for the error path detector [22]. In Fig. 6(b), (c), similar results are observed for the IEEE802.11n LDPC and 5G LDPC codes, where the proposed scheme outperforms the other schemes and achieves an FER of $10^{-7}$ without a severe error-floor.

## 5   Conclusions

This paper proposed training methods for the NMS decoder of LDPC codes to enhance the error-floor performance. Using uncorrected words from the base decoder, we trained the post decoder to be specialized for error patterns causing the error-floor, promoting decoding diversity in the cascaded base and post decoders. We also proposed a training schedule to circumvent the vanishing gradient and local minimum problems, and a weight sharing technique that significantly reduces the number of distinct weights without sacrificing performance. The proposed NMS decoder using the trained weights showed the excellent waterfall and error-floor performances for several standard LDPC codes. Along with the performance improvement, the proposed training scheme has the advantage of being flexibly applicable regardless of the types of channel, code, and decoding algorithm. This scheme can also be implemented directly on hardware architectures without additional costs, and can be directly utilized with no prior analysis of the target code and decoding algorithm.

## 6   Acknowledgments

This work was supported by Samsung Electronics Co., Ltd (IO230411-05859-01), by Electronics and Telecommunications Research Institute (ETRI) grant funded by the Korean government [2021-0-00746, Development of Tbps wireless communication technology], by the National Research Foundation of Korea (NRF) grant funded by the Korea government (MSIT) (No. RS-2023-00212103),

and by the National Research Foundation of Korea (NRF) grant funded by the Korea government (MSIT) (No. RS-2023-00247197).

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

# 7 Appendix

## 7.1 A better trade-off between the waterfall and error-floor performances

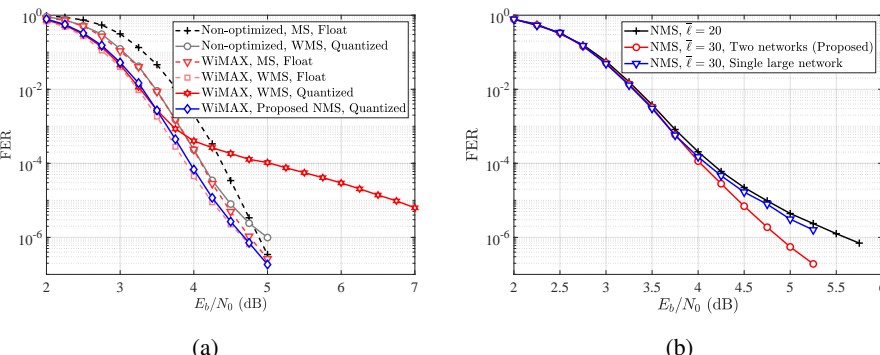

Figure 7: (a): Decoding performance with optimized/non-optimized codes and floating/quanztized decoders (b): Comparison between the two networks and single large network.

Typically, LDPC code design assumes the availability of a floating-point decoder and optimizes the code to maximize waterfall performance. In Fig. 7(a), it's evident that the optimized WiMAX LDPC code outperforms the non-optimized regular LDPC code in terms of the waterfall performance when the floating-point MS decoder is employed. Additionally, the WMS decoder enhances the performance by assigning weights. As shown in Fig. 7(a), WMS decoding of the WiMAX LDPC code exhibits the best waterfall performance when floating-point operations are available. However, for practical hardware implementation, quantization of decoding messages is essential, but it leads to performance degradation. Notably, WMS decoding suffers substantial performance degradation with an early onset of the error-floor. A side-by-side comparison of the quantized WMS decoding performances between the non-optimized and WiMAX LDPC codes reveals that the performance loss due to quantization is more pronounced in optimized LDPC codes. Consequently, there's a need for a method that ensures high performance in both the waterfall and error-floor regions with low complexity. This work accomplishes this goal by leveraging machine learning techniques. In Fig. 7(a), the proposed quantized NMS decoder achieves the floating WMS decoding performance for the WiMAX LDPC code in both waterfall and error-floor regions.

## 7.2 Boosting learning of two networks v.s. conventional learning of a single large network

In Fig. 7(b), we compare the proposed method with the method that uses a single decoder, which is trained by a large single network at once. For the proposed method, the boosting learning is employed with two networks. Both have the same model size. Fig. 7(b) shows that simply increasing the model size doesn't significantly mitigate the error-floor. In contrast, the two-stage network using boosting learning achieves greater decoding diversity and demonstrates superior efficiency in alleviating the error-floor.

## 7.3 Optimization of parameters of the block-wise training schedule

Table 3: Test FER values of the block-wise training with various $\Delta_1$ and $\Delta_2$.

| $\Delta_1 \setminus \Delta_2$ | 0 | 5 | 10 | 15 | 30 |
|---|---|---|---|---|---|
| 1 | 0.376 | 0.144 | 0.123 | 0.122 | 0.150 |
| 5 | 0.193 | 0.134 | **0.112** | 0.127 | 0.137 |
| 10 | 0.142 | 0.142 | 0.13 | 0.128 | 0.13 |
| 30 | 0.179 | | | | |

To fine-tune the parameters $\Delta_1$ and $\Delta_2$ used in the block-wise training schedule, we compare the test FER values across a range of $\Delta_1$ and $\Delta_2$. The comparison is done for the post decoder training

with $\ell_2 = 30$ and is summarized in Table 3. For the case where $\Delta_2 = 0$ (i.e., without retraining), the performance improves as $\Delta_1$ increases from 1 to 10. This can be attributed to the training of more weights simultaneously, aiding in escaping from local minima. However, if $\Delta_1$ becomes too large, the weights in front are not adequately trained due to the vanishing gradient problem, causing performance degradation. In addition, as the retraining iteration number $\Delta_2$ grows for a given $\Delta_1$ value, a similar trend is noted: performance enhances up to a certain point and then plateaus. According to Table 3, the best performance is achieved with $\Delta_1 = 5$ and $\Delta_2 = 10$.

## 7.4 Discussion on the trained weights

Table 4: Trained weights

| | $\ell$ | Weights | | | | | | | | | |
|---|---|---|---|---|---|---|---|---|---|---|---|
| VW $\overline{w}^{(\ell)}$ | $1 \sim 10$ | 0.74 | 0.98 | 0.96 | 1.10 | 1.20 | 1.15 | 1.22 | 1.19 | 1.17 | 1.09 |
| | $11 \sim 20$ | 1.15 | 1.09 | 1.07 | 1.04 | 1.05 | 1.07 | 1.00 | 0.98 | 0.90 | 0.81 |
| | $21 \sim 30$ | 1.14 | 0.97 | 0.93 | 0.69 | 0.66 | 0.58 | 0.57 | 0.51 | 0.49 | 0.45 |
| | $31 \sim 40$ | 0.48 | 0.41 | 0.37 | 0.38 | 0.38 | 0.39 | 0.32 | 0.33 | 0.32 | 0.34 |
| | $41 \sim 50$ | 0.39 | 0.37 | 0.31 | 0.31 | 0.34 | 0.45 | 0.39 | 0.37 | 0.42 | 0.42 |
| CW $w^{(\ell)}$ | $1 \sim 10$ | 0.74 | 0.71 | 0.66 | 0.69 | 0.67 | 0.70 | 0.75 | 0.71 | 0.73 | 0.76 |
| | $11 \sim 20$ | 0.73 | 0.70 | 0.77 | 0.79 | 0.84 | 0.84 | 0.86 | 0.82 | 0.81 | 0.97 |
| | $21 \sim 30$ | 0.19 | 0.30 | 0.54 | 0.58 | 0.62 | 0.59 | 0.73 | 0.73 | 0.74 | 0.78 |
| | $31 \sim 40$ | 0.72 | 0.78 | 0.79 | 0.81 | 0.88 | 0.82 | 0.88 | 0.85 | 0.97 | 0.79 |
| | $41 \sim 50$ | 0.81 | 0.82 | 0.85 | 0.91 | 1.19 | 0.86 | 0.96 | 1.10 | 1.18 | 1.62 |
| UCW $\hat{w}^{(\ell)}$ | $1 \sim 10$ | 0.74 | 0.71 | 0.66 | 0.69 | 0.67 | 0.70 | 0.75 | 0.71 | 0.73 | 0.76 |
| | $11 \sim 20$ | 0.73 | 0.70 | 0.77 | 0.79 | 0.84 | 0.84 | 0.86 | 0.82 | 0.81 | 0.97 |
| | $21 \sim 30$ | 0.58 | 0.80 | 0.83 | 0.73 | 0.81 | 0.86 | 0.72 | 0.78 | 0.73 | 0.73 |
| | $31 \sim 40$ | 0.59 | 0.71 | 0.64 | 0.67 | 0.67 | 0.72 | 0.65 | 0.65 | 0.66 | 0.70 |
| | $41 \sim 50$ | 0.77 | 0.70 | 0.72 | 0.81 | 0.92 | 0.58 | 0.71 | 0.76 | 0.87 | 1.57 |

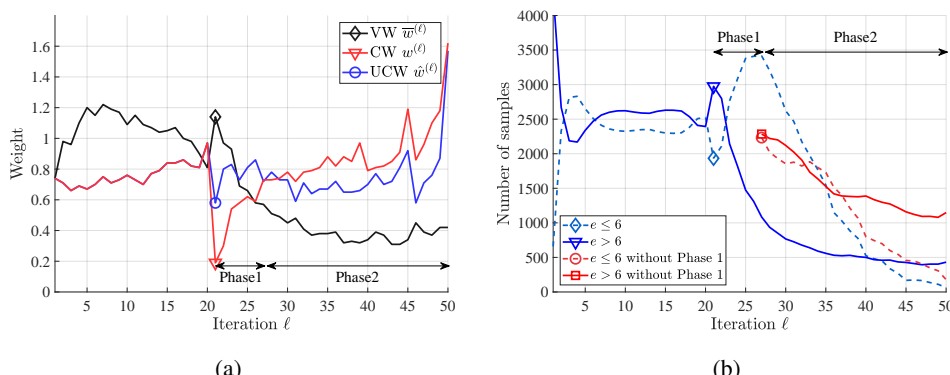

|       |       |
|-------|-------|
| (a)   | (b)   |

Figure 8: Evolution of the weights and the number of samples as a function of iteration.

Table 4 shows the resulting weights trained using the proposed training methods. Owing to the weight sharing technique, only three types of weights, VW $\overline{w}^{(\ell)}$, CW $w^{(\ell)}$, and UCW $\hat{w}^{(\ell)}$, exist for each iteration. Thus, weight variations across iterations can be depicted in two dimensions, as shown in Fig. 8(a), enabling analysis of the trained results. Without the weight sharing technique, there would be $(N + E)$ weights per iteration, making a 2D graphical representation unfeasible. Fig. 8(a) shows that the weights for the base decoding stage, where the CW and VW are roughly $0.75$ and $1$, are similar to the weights of conventional WMS decoding. As a result, the waterfall performance of WMS decoding and NMS decoding are nearly indistinguishable. The NMS decoding technique offers minimal advantage in the waterfall region.

On the other hand, for the post decoding stage, the weights undergo significant changes. At iteration 21, the VW increases while the CW decreases substantially. This means that the channel LLR values are given more weight when performing the sum operation at each VN, while the messages coming from CNs are attenuated. As a result, the decoding process reverts to the initial decoding state (with

a high error rate). This leads to an increase in the number of samples having more than 6 errors (denoted by $e > 6$), as shown in Fig. 8(b), and a decrease in the number of samples with $e \leq 6$. This intentional regression aims to break free from trapped error patterns. From iteration 22 onward, the CW gradually increases, the VW decreases, and the count of $e \leq 6$ samples starts to increase again. After iteration 27, the CW and VW show less variations. We term the period from iteration 21 to 26 as Phase 1 and the period following iterations as Phase2. At iteration 20, the numbers of samples with $e \leq 6$ and $e > 6$ are roughly equal, but through Phase 1, samples with $e > 6$ transform into samples with $e \leq 6$. Consequently, by the end of Phase 1, there are more samples with $e \leq 6$ than samples with $e > 6$. The small-sized error samples diminished steadily in Phase 2.

The result without Phase 1 is also shown in Fig. 8(b). Without Phase 1, the process of transforming into small-sized error samples is omitted. Consequently, the decoding proceeds in a situation where the number of samples with $e \leq 6$ and $e > 6$ is similar. Although Phase 2 effectively correct most of the $e \leq 6$ samples, the $e > 6$ samples remain in significant numbers as they are initially abundant. This indicates the need for the pre-processing of Phase 1.

## 7.5    Multiple stage decoders

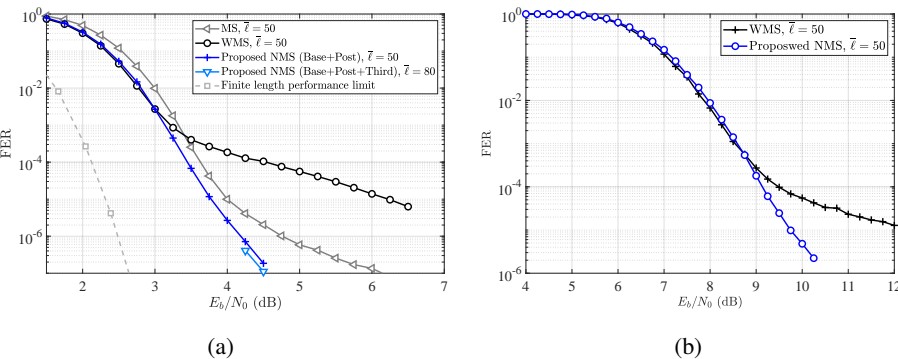

Figure 9: (a): FER performance of the triple decoder with the theoretical limit for the WiMAX LDPC code (b): FER performance over the Rayleigh channel for the WiMAX LDPC code.

Extending the proposed method to include more than two decoders is feasible and can be applied in a similar manner. For instance, in the case of the WiMax LDPC code (shown in Fig. 6(a)), uncorrected words from the error-floor region of the base + post decoder (i.e., $E_b/N_0$ 4.5 dB) are collected to train a third decoder. Fig. 9(a) shows that the aid of this third decoder (labeled as "Base+Post+Third) reduces the error-floor further. It is worth mentioning that introducing this third decoder might lead to increased decoding latency. Moreover, the process of collecting uncorrected words would be time-consuming, especially in the very low FER region. Addressing these challenges could be a promising direction for subsequent research. In addition, we add a graph showing the finite length performance limit [44].

We want to note that the concept of multi-stage decoding has been employed in the previous works [45, 46, 47], particularly in the field of coding theory. Our study has similarities with theses previous approaches, as we also carries out a two-stage decoding. However, unlike physically divided decoders in [45, 46, 47], our method utilizes a single LDPC decoder. While the proposed decoder is conceptually divided into two stages (base/post) based on iterations, in practice, we employ only a single decoder with distinct weight parameter sets. Moreover, our approach has a novelty in that the post decoder is 'trained' dependent on the results of the base decoder.

## 7.6    Application to a different channel

The concept of training using uncorrected words can be consistently applied regardless of the channel type. In Fig. 9(b), we include the results for the Rayleigh channel with the scale parameter $\alpha = 1$, showing the effectiveness of our proposed method holds across different channel types.

## 7.7 Application to long length codes

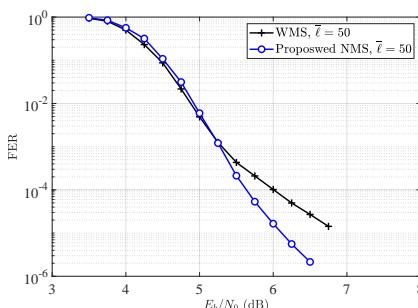

Figure 10: FER performance for the long 5G LDPC code of length 1248.

We show the results for the (1248, 1056) 5G LDPC code in Fig. 10. This result demonstrates the proposed method is also effective for long codes. There is little improvement in waterfall performance with the NMS technique at such long lengths. However, the performance improvement in the error-floor region using our proposed method is clearly evident.

## 7.8 Comparison with the augmented neural decoders

Table 5: Comparison with [5], [34], [6] in terms of the waterfall performance of short codes. The results are measured by the negative natural logarithm of BER. Higher is better.

| Architecture | | | | Vanilla NBP | | | | | | NBP+HyperNet. | | | Transformer | | |
|---|---|---|---|---|---|---|---|---|---|---|---|---|---|---|---|
| Method | | BP | | | Orig. NBP [5] | | | Proposed | | | Hyper [34] | | | ECCT [6] | |
| $E_b/N_0$ | 4 | 5 | 6 | 4 | 5 | 6 | 4 | 5 | 6 | 4 | 5 | 6 | 4 | 5 | 6 |
| Polar (64,48) | 4.74 | 5.94 | 7.42 | 4.70 | 5.93 | 7.55 | 4.93 | 6.64 | 8.77 | 4.92 | 6.44 | 8.39 | 6.36 | 8.46 | 11.09 |
| BCH (63,51) | 4.58 | 5.82 | 7.42 | 4.64 | 6.21 | 8.21 | 4.72 | 6.42 | 8.96 | 4.80 | 6.44 | 8.58 | 5.66 | 7.89 | 11.01 |
| MacKay (96,48) | 8.15 | 11.29 | 14.29 | 8.66 | 11.52 | 14.32 | 8.26 | 11.83 | 15.85 | 8.90 | 11.97 | 14.94 | 8.39 | 12.24 | 16.41 |

In Table 5, we compare the waterfall performance of various neural decoders for short length codes. To ensure a fair comparison with other works, we employ the NBP decoder and utilize the soft-BER loss function for BER performance optimization while making use of full diversity weights. Out of a total of 50 iterations, 20 iterations are allocated for base decoding and 30 iterations for post decoding.

The result reveals a notable performance improvement over the original NBP [5], which employs the same vanilla NBP architecture. The improvement is particularly pronounced at high SNR. We achieve even better performance at high SNR than the work [34], which adds hyper-networks to the vanilla NBP architecture. In comparison to the ECCT [6] using the transformer architecture, our method shows inferior performance for high density codes (Polar, BCH), yet it achieves fairly comparable performance for LDPC codes. Owing to its intricate transformer structure, the ECCT requires higher training and decoding complexity than NMS decoders [6], implying its current limited practicality. Nevertheless, it is worth emphasizing that our proposed training method can be adaptable across any architecture. In other words, there's promising potential in future research that integrates the proposed training methods with the architecture with hyper-networks or the transformer architecture.

## 7.9 Ablation study

Table 6: Ablation analysis.

| O: Boosting X: Conv. | O: Block-wise X: One-shot | O: Proposed sharing X: Full diversity | FER (at $E_b/N_0$ 5.0dB) |
|---|---|---|---|
| X | X | X | $3.14 \times 10^{-6}$ |
| O | X | X | $2.77 \times 10^{-7}$ |
| O | O | X | $1.84 \times 10^{-7}$ |
| O | O | O | $1.85 \times 10^{-7}$ |

In Table 6, we provide the ablation study for the WiMAX LDPC code. The result shows that the boosting learning is the core technique to reduce the FER performance and the block-wise training also contributes the FER reduction, while the proposed sharing technique does not involve performance degradation.

