# OpenReview forum: "Boosting Learning for LDPC Codes to Improve the Error-Floor Performance"
_NeurIPS.cc/2023/Conference — NeurIPS 2023 poster_

### Official Review · Reviewer_UbgL · 2023-06-17

**Soundness:** 3 good
**Presentation:** 2 fair
**Contribution:** 3 good
**Rating:** 7
**Confidence:** 5

**Summary:**


1.	In the proposed work authors proposed
         a.	Neural Min-Sum (NMS) decoders
        b.	NMS decoder with block-wise training schedule that locally trains a block of weights while retraining the preceding block.
        c.	The different weights are assigned to the unsatisfied check nodes and training is carried out.
2.	The contribution of the proposed work are.
     a.	Boosting learning using uncorrected codewords
     b.	Block-wise training schedule with retraining
     c.	Weight sharing technique with dynamic weight allocation
3.	Sufficient results are presented and discussed.


**Strengths:**

Novelty of the work is good.
Sufficient results are presented and discussed.

**Weaknesses:**

The experimetnal set up should be elaboprated more with parameters selected for training.

**Questions:**

1.	How the weights of the NMS decoders are initialized.
2.	Is it possible to design ASIC hardware of NMS decoder
3.	What about the latency of the decoder.
4.	The neural decoders with recent state of art techniques with their advantages should be discussed in detail.
5.	The experimental set up should be clearly mentioned with parameters used for the training.
6.	How exactly the weights are updated in the training process of LDPC CODES mentioned in the figure 1 and what is the impact of weights in the training process should be clearly mentioned.


**Limitations:**

Since the authors mentioned that proposed work can be taken into hardware architecture.
But clear architectute of the proposed method is not highlighted.

---

> ### Author Rebuttal · Authors · 2023-08-09
>
> Thank you for your positive feedback and constructive comments.
>
> ## Experimental setup
> As suggested, we’ve provided a more detailed account of the experimental setup in the revised version, including the channel type, the training Eb/N0 for the post decoder for each code, the number of hidden layers, and the number of nodes in the neural network.
>
> ## Initial values
> Initial values are set to 1. We’ve tried various values, but there wasn't a significant difference. Research on effective initialization is an interesting topic, and we plan to pursue it in follow-up studies.
>
> ## Implementation by ASIC
> The proposed method requires no additional modules; hence, once trained, the NMS decoder can be implemented in the same manner as the WMS decoder. There have been numerous studies [a]-[c] on the ASIC implementation of the WMS decoder. As a result, the proposed NMS decoder can also be fabricated using ASIC hardware.
>
> [a] W. Zhang, S. Chen, X. Bai, and D. Zhou, “A full layer parallel QCLDPC decoder for WiMAX and Wi-Fi,” in Proc. IEEE 11th Int. Conf. ASIC (ASICON), Nov. 2015, pp. 1–4.
>
> [b] T. H. Tran, Y. Nagao, H. Ochi, and M. Kurosaki, “ASIC design of 7.7 Gbps multi-mode LDPC decoder for IEEE 802.11ac,” in Proc. IEEE 14th Int. Symp. Commun. Inf. Technol. (ISCIT), 2014, pp. 259–263.
>
> [c] S. Shao et al, “Survey of Turbo, LDPC, and Polar Decoder ASIC Implementations”,  IEEE Communications surveys & Tutorials, Vol. 21, no. 3, pp. 2309-2333, 2019
>
> ## Comparison with recent works
> Conventional neural decoder works have primarily targeted improvements in waterfall performance, whereas our study achieved improvements in error floor performance. Nevertheless, we include a comparison with the latest neural decoder study [15] in the attached file’s Table R.1. Compared to [15], our method achieves similar performance with lower training/decoding complexity.
>
> A comparison in the error floor region is not able to conduct with [15], as verifying their error floor performance is infeasible, and there are numerous differences in the experimental environments. The state-of-the-art (SOTA) result with the same experimental setting with our work is from [33], so we performed the comparison with that. The distinctions between our research and these prior works will be highlighted in the revised version.
>
>
> ## How to update the weights
> Figure 1(b) corresponds to the neural network for the LDPC code depicted in Figure 1(a). The connections between the nodes in the network vary depending on the code, and each edge has an assigned weight. Here, these weights do not have any special functions; they are just typical weights of neural networks. Therefore, the training of these weights is done using standard backpropagation, and in this paper, we employed the Adam optimizer.
>
> ## HW architecture
> It seems that some ambiguous expressions may have led to misunderstandings. What we meant was that the proposed method can be directly applied to the HW architecture used in WMS, not that we are proposing a new architecture. We will make corrections to the revised version to ensure clear communication of this point.

---

### Official Review · Reviewer_jBEt · 2023-06-21

**Soundness:** 3 good
**Presentation:** 2 fair
**Contribution:** 3 good
**Rating:** 6
**Confidence:** 4

**Summary:**

The paper proposes a training framework for the NMS decoder of LDPC codes in order to enhance the error-floor performance of these codes. \
The NMS decoder iterations are split into two cascaded parts, the first so-called based decoder is trained for decoding waterfall parts, and the second (post decoder) is for decoding uncorrected codewords caused by the error-floor.\
The paper suggests the use of a training schedule reusing learned weights to tackle the vanishing gradient of many iterations NMS.\
The proposed decoder using the trained weights showed improved error-floor performances for several standard LDPC codes and can be efficiently integrated into current NMS solutions.

**Strengths:**

The paper addresses an important issue present in BP/MS-based decoders on these codes.

The solution is simple and does not require any architectural modification.

The performance seems important and is obtained without adding any complexity compared to the original NMS decoder.

**Weaknesses:**

#### NMS
 I am not sure the comparison with NMS is fair for multiple reasons.

1) The training of NMS can be performed in several ways that should not allow zero loss. For example, the observations in lines 224 and 225 can be easily solved by using large batches and/or using batches spanning many Eb/N0 values.


2) The vanishing gradient phenomenon has been solved in [5] using a multi-loss objective, which clearly prevents the gradients to collapse at each layer.
Also, in Figure 4b) NMS and the method should be compared at equal number of iterations.
NMS's test FERs with l=40 and l=50 are not that far from the  l=50 proposed method FER.

3) NMS remains non-negligibly better in the waterfall region.

4) Many NMS algorithms have emerged since [5] and I am wondering if the comparison with the original NMS [5](2018) is fair and conclusive in order to judge the paper.

#### Clarity:
Figure 2 is not clear. For example, the differentiation between the NMS and the neural network is not clear; the base and post NN should be linked by an arrow.

The paper contains a very large amount of text which can become cumbersome. I believe adding clear Algorithm(s) would simplify the comprehension a lot.


#### Misc:
Typos:

line 88: we a new \
line 121: protogrph

**Questions:**

1) Please address the NMS subsection from the Weaknesses section.\
It would be beneficial to have an ablation study of the final accuracy for NMS augmented with the different components of the proposed methods. Also, different training strategies of the NMS must be considered.

2) Please address the clarity and Misc from the Weaknesses section.

I believe adding clarity, addressing the ablations, and adding more experiments would increase the quality, impact, and rating of the paper.

**Limitations:**

No limitations are discussed.

---

> ### Author Rebuttal · Authors · 2023-08-09
>
> Thank you for providing insightful comments.
>
> ## Schemes for avoiding zero loss
> Thank you for your detailed feedback. Lines 224, 225 are about the Case 3. In Case 3, we use received words sampled at 4.5dB (the error floor region of the base decoder) as training samples for the post decoder without any filtering. At 4.5 dB, the FER of the base decoder is at the level of $2\times 10^{-5}$, so even if the batch size is increased to 100, on average, only 0.02 samples produce a non-zero loss. Furthermore, if we span the sampling Eb/N0 to include lower values (in the Waterfall region), it doesn't foster the post decoder training that's robust to the error floor. In other words, the core idea of our proposed method is to sample from the error floor region and use only uncorrected words by filtering, ensuring effective training for the post decoder. We will refine this section to eliminate any ambiguity and preclude potential misinterpretations.
>
> ## Multi-loss objective function
> As you suggested, we included a comparison with the multi-loss method (see the attached file, Figure R.1(d)). Like the iter-by-iter method, the multi-loss method also rapidly reduces the FER value in the initial iterations. However, by iteration 50, block-wise training outperforms the multi-loss method. This can be attributed to the fact that error patterns occurred in the error floor region can be better corrected when all iterations within a block collaborate closely, rather than optimizing each individual iteration. This is discussed in Section 4 of the supplementary material. While the graph in Figure R.1(d) seems not to show a significant difference in the final test FER, the precise values are 0.179 for at once, 0.156 for multi-loss, and 0.112 for block-wise, which means that the block-wise training method achieves reductions of 40% and 28% respectively.
>
> ## NMS remains non-negligibly better in the waterfall region
> The proposed post decoder is trained for the error floor region, so even with the addition of the post decoder, the waterfall performance remains largely unchanged. (compare the proposed method with iteration 20 and 50 in Figure 6(a)) Consequently, as you rightly pointed out, our proposed method in Figure 6(a) exhibits a slightly degraded waterfall performance compared to other NMS methods. Compared to [5], our methodology incurs a minor reduction of 0.05dB in the waterfall region (@FER $10^{-2}$). However, it can achieve an appreciable SNR gain of 1dB in the error floor region (@FER $10^{-7}$). Such a trade-off will be beneficial, especially for the target applications of our work, e.g., extremely ultra-reliable communications for 6G and storage systems.
>
> ## Comparison with other works
> As you've pointed out, several follow-up studies have been conducted after [5]. Unlike our work focuses on the error floor performance, the majority of the works have concentrated on improving waterfall performance. The state-of-the-art (SOTA) model-based approaches in waterfall performance is the hyper graph network [15]. A comparison with [15] in the waterfall region is added to the attached file. Our proposed boosting learning is also applicable in the waterfall region. As seen in Table R.1, our method achieves comparable performance with [15] solely through the enhancement of the training technique without any additional modules.
>
> What we would like to emphasize is that our proposed method has achieved the SOTA result in terms of the error floor performance under a low complexity environment. Specifically, our work assumes scenarios suitable for application areas like storage systems and XURLLC for 6G, encompassing:
>
> i) Error floor region (FER <=1e-7, BER<=1e-9)
>
> ii) Target metric: FER performance
>
> iii) MS algorithm
>
> iv) Without additional training/decoding cost
>
> v) Hardware-Feasible
>
> vi) Moderate and long code length (>=500)
>
> In contrast, [15] targets short-length BER performance in the waterfall region (BER~1e-7), incurs additional training/decoding costs, and is constrained to the BP algorithm. This makes it difficult to make a fair comparison. Technically, due to memory limitations in TensorFlow, it was impossible for us to train the [15] technique on the WiMax LDPC code of length 576. Moreover, the evaluation of the technique [15] in the waterfall region takes several days and it is infeasible to evaluate the error-floor performance.
>
> Our proposed method is the first in research on learning-based decoders to directly target the error floor performance. For reference, the SOTA work sharing the same environment as this paper is [33], and our proposed NMS showed better performance with about a third of the latency and without additional modules. In the revised paper, we will emphasize that the SOTA in the waterfall is the existing works [15], and will highlight that the goal of our study is the error floor. Thank you for clarifying the contribution of the research.
>
> ## Clarity and ablation study
> Thank you for the valuable suggestion. For the clarity of the paper, we've extensively revised the schematics as shown in Figure R.5 of the attached file and added an algorithm. Additionally, we included an ablation study in the attached file. The result shows that the boosting learning is the core technique to reduce the FER performance and the block-wise training also contributes the FER reduction, while the proposed sharing does not involve the performance degradation.

---

> > ### Comment · Reviewer_jBEt · 2023-08-10
> >
> > Thank you for your very clear and thorough answer.
> >
> > 1) Thank you for the experiment. What is the impact of multi-loss applied with your method? Do you assume it won't help?
> >
> > 2) Comparison with other works: the comparison is not very pertinent since much better neural decoders than [15] have been recently developed (e.g., [A, B] are related to the authors of [5] and [15]). However, I admit their capacities may not be straightforwardly equivalent (e.g., [C] is not performing as well but is a more lightweight model).
> >
> > 3) It seems the careful selection of the data solely seems to have an impressive and surprising impact on the performance.
> >  - Do you know what is the real impact of training and fine-tuning on the whole model (base+post)? (maybe with different step-sizes)
> >  - Do you know if other data-selection policies during training have already been investigated? (related works focus on weights repartitions).
> >
> > Thank you
> >
> > [A] *Autoregressive Belief Propagation for Decoding Block Codes*
> >
> > [B] *Error Correction Code Transformer*
> >
> > [C] *Graph Neural Networks for Channel Decoding*

---

> > > ### Author Response · Authors · 2023-08-13
> > > **Reply to the comments by jBEt**
> > >
> > > 1.
> > >
> > > In the error floor scenario with FER metric, the proposed block-wise method shows the best results, but the multi-loss method is the second most effective. The multi-loss approach has the advantage of enabling learning over a large number of iterations without suffering from the vanishing gradient problem unlike the at-once method. Also, in contrast to the iter-by-iter method, it incorporates multiple iterations in the loss function, which more effectively eliminates error patterns in the error floor. As a result, the multi-loss method achieves a lower FER compared to the at-once and iter-by-iter methods. We will mention these advantages in the revised version.
> > >
> > > 2.
> > >
> > > Thank you for the valuable comment. We will add a comparison with other works [A], [B]. (A comparison with [C] is not conducted as the source code is not publicly accessible and an apple-to-apple comparison is little hard) As is known, in the waterfall region, the performance of [B] demonstrates state-of-the-art results. Especially for high-density codes such as BCH and polar codes, there's a significant performance gap when compared with BP and MS research series (Hyper [15], AR [A], Boosting [Proposed]), whereas the gap is relatively minimal for LDPC codes.
> > >
> > > However, [B] uses an entirely different architecture based on transformer, which is not based on BP and MS algorithms, making its practicality somewhat limited as of now. Additionally, due to its intricate network structure, it's more complex than NMS decoders in terms of training time, memory requirement, and decoding complexity. In practice, the ECCT has difficulty training for the target codes of this paper, particularly for codes of several hundred lengths with very low error rates. In our GPU environment, training for the WiMax code (N=576, K=432) is feasible using only `shallow` ECCT (N=2, d=32), and at this depth, there's no performance improvement over the BP & MS series of works. On the other hand, our proposed boosting learning is trainable even in very deep networks, around 100 iterations, and long codes 1000 in length.
> > >
> > > Although our focus in this work was on improving the error floor performance of neural min-sum decoders, our approach could potentially be applied to other decoding architectures (e.g., ECCT). Application of our boosting learning to ECCT would be an enticing follow-up as a future study. We conjecture that the boosting learning via smart selection and arrangement of training samples can be effective on a large class of architectures.
> > >
> > > In summary, the contributions of our work are as follows:
> > >
> > > i) Primary contribution: Achieving SOTA in terms of error floor performance in practical situations like the NMS decoder, standard LDPC codes.
> > >
> > > ii) Secondary contribution: Achieving a competitive waterfall performance for short codes by modifying only the training method. (will be discussed in the supplementary file)
> > >
> > > iii) Additional benefit: A flexible methodology applicable to different channels and decoder architectures.
> > >
> > > 3.
> > >
> > > As you mentioned, careful selection of data samples can lead to a significant improvement in decoding performance through the proposed boosting approach. The ablation study underscores this substantial performance boost (as seen when comparing the first and second rows of the ablation study: from $3.14 \times 10^{-6}$ to $3.31 \times 10^{-7}$.
> > >
> > > Most research has focused on proposing new decoder architectures or decoding algorithms to improve decoding performance. In contrast, our proposed boosting approach demonstrates that the decoding performance can be significantly enhanced by training carefully selected data samples. We believe that the proposed boosting approach offers a promising new research direction that can improve decoding performance while minimizing the additional decoding complexity.
> > >
> > > I've explored fine-tuning of some hyper-parameters (learning rate, initial values, batch size) but didn't observe significant effects. Instead, we will include experimental results showing valid effects, such as the usage of VN weights and splitting of the number of iterations between base and post.
> > >
> > > Research on data selection has been conducted in “active learning” [16] and “BP-RNN diversity” [9]. [16] employs a single-stage decoding and gathers data samples with a moderate number of errors, inspired by active learning. However, [16] significantly differs from our proposed boosting learning method, which uses a two-stage decoding process and collects the failed codeword samples during the first stage. BP-RNN Diversity [9] requires enumerating trapping sets beforehand and because of this, it can only be applied to short codes. In contrast, the sampling of uncorrected words in the proposed boosting learning has an O(n) complexity, making it feasible for longer codes. More importantly, they focused on the waterfall performance. We will clarify this point in the revised manuscript. Thank you again for the valuable comments.

---

> > > > ### Comment · Reviewer_jBEt · 2023-08-16
> > > >
> > > > Thank you for your answer. I believe the rebuttal improved the paper and even if the neural contribution is not substantial, the problem and the simple solution may be impactful enough.

---

### Official Review · Reviewer_MFD6 · 2023-07-04

**Soundness:** 3 good
**Presentation:** 4 excellent
**Contribution:** 3 good
**Rating:** 6
**Confidence:** 4

**Summary:**

The author studies LDPC code's neural min-sum decoder's error floor problem, by solely change the training method:

(1) boosting learning method, which use first network to do majority of decoding to achieve waterfall region, and second network deal with small error residuals.
(2) relief deep decoding iteration's vanish gradient problem by block-wise training.
(3) bundle weight in dynamic manner, to reduce the number of weights to be trained.

With above techniques, the experiment runs on varies LDPC settings, shows that proposed NMS has better performance on error-floor regions, with same computational resource as NMS. Since NMS and MS have similar complexity, it is believed that proposed NMS could be possibly adapted to future LDPC decoder implementation.

**Strengths:**

(1) The paper works directly on training of NMS algorithm, which requires minimal change on existing low complexity NMS solutions. All three training methods are reasonable and are supported by experiments. The residual learning (termed as boosting learning) empirically shows advantage on error-floor regions. The block-wise training schedule does improve performance with large number of iterations. Finally the weight sharing technique adds a structural sparse limitation to neural decoder, which only uses a small portion of weights.

(2) The evaluation is extensive, on multiple LDPC settings at Figure 6, we see that proposed NMS method shows better error-floor performance, compared to existing NMS and canonical decoders.


**Weaknesses:**

Here are my concerns:

(1) The major contribution on Machine Learning sense, is limited. The proposed boosting learning method is common, as residual learning in ML sense; the proposed training schedule and weight sharing are also widely-used ML methods. The core contribution is on applying ML techniques in the context of channel coding field. In some sense, the targeted audience shall be channel coding researchers, rather than general ML researchers.

(2) The experiments are not conducted in long block length (length>1000, which is common for QC-LDPC), which is widely used in 5G/WIFI systems. It might be interesting to check the performance on long block length, where block length coding gain is significant, and NMS shows less advantage against canonical methods.

**Questions:**

N/A

---

> ### Author Rebuttal · Authors · 2023-08-09
>
> Thank you for your valuable comment.
>
> ## Contribution of this work
> We agree with your opinion that the contribution to machine learning techniques can be seen as limited. However, we believe this manuscript aligns well with the scope of NeurIPS under the category of “application of machine learning”. The primary emphasis of our work is on the novel application of ML techniques to the domain of coding theory. In a similar vein, previous works [a]-[d], presented at ML conferences, have primarily aimed at broadening the spectrum of ML applications. A unique aspect of our research is the adoption of boosting learning, a method yet to be explored in coding theory. Our boosting approach for LDPC codes achieves the SOTA performance in the error-floor region compared to other LDPC decoding strategies.
>
> [a] 2018 NeurIPS, “Deepcode Feedback Codes via Deep Learning”
>
> [b] 2019 NeurIPS, “Hyper-Graph-Network Decoders for Block Codes”
>
> [c] 2020 NeurIPS, “Learning to Decode Reinforcement Learning for Decoding of Sparse Graph-Based Channel Codes”
>
> [d] 2021 ICML, “Cyclically Equivariant Neural Decoders for Cyclic Codes”
>
> [e] 2022 NeurIPS, “Error Correction Code Transformer”
>
> ## Long length code
> As you suggested, we’ve added the results for the (1248, 1056) 5G LDPC code to the attached file's Figure R.1(c). This result demonstrates the proposed method is also effective for long codes. As you mentioned, there is little improvement in waterfall performance with the NMS technique at such long lengths. However, the performance improvement in the error floor region using our proposed method is clearly evident.

---

> > ### Comment · Reviewer_MFD6 · 2023-08-18
> > **Keep my score the same**
> >
> > After reading the response from the author, I decide to keep my rating.

---

### Official Review · Reviewer_zXhE · 2023-07-06

**Soundness:** 4 excellent
**Presentation:** 4 excellent
**Contribution:** 4 excellent
**Rating:** 7
**Confidence:** 5

**Summary:**

The paper proposes training methods to optimize neural min-sum (NMS) decoders that are robust to the error-floor phenomenon of LDPC codes. The proposed methods include: (1) dividing the decoding network into two neural networks and training the post network to be specialized for uncorrected codewords that failed in the first network; (2) introducing a block-wise training schedule that locally trains a block of weights while retraining the preceding block; and (3) assigning different weights to unsatisfied check nodes. The proposed methods are applied to standard LDPC codes, and the results show that they achieve the best decoding performance in the error-floor region compared to other decoding methods in the literature. The proposed NMS decoder is designed only by modifying the training methodology, without adding any additional modules. Therefore, it can be seamlessly incorporated into the well-established architectures of LDPC decoders and can be immediately utilized in practical domains that demand exceptionally low error rates.



**Strengths:**

The paper introduce novel ideas:
1. Boosting learning using uncorrected codewords - The authors propose a boosting learning technique for LDPC code decoding. The technique divides the decoding network into two cascaded networks, where the first network focuses on the waterfall performance and the second network focuses on the error-floor region. The second network is able to correct uncorrected codewords that are not corrected by the first network due to the error-floor phenomenon. This results in a significant performance improvement in the error-floor region.
2. Block-wise training schedule with retraining - The authors propose a new training schedule for NMS decoders that mitigates the vanishing gradient problem. The proposed schedule divides the entire decoding iterations into sub-blocks and trains the blocks in a sequential manner. Additionally, the weights trained from previous blocks are retrained to escape from local minima. This results in a significant performance improvement over the one-shot training method and the iter-by-iter schedule.
3. Weight sharing technique with dynamic weight allocation - The authors propose a new weight sharing technique for NMS decoders that improves the performance in the error-floor region. The technique dynamically assigns different weights to unsatisfied check nodes (UCNs) and satisfied check nodes (SCNs) in the decoding process. This results in a significant performance improvement with a 2.6% reduction in the number of weights to be trained.


**Weaknesses:**

.

**Questions:**

1. Does the proposed method can be also beneficial to improve the results of neural decoder for other types of codes? such as BCH and Polar?
2. How does the model behave with other type of noise (other than Gaussian)?
3. Can you tie the weights of the different network to get improvement in terms of model size?
4. What happen if you take more then two NN? does the performance improve?

**Limitations:**

.

---

> ### Author Rebuttal · Authors · 2023-08-09
>
> Thank you for your positive feedback and constructive comments.
>
> ## Other types of codes
> The proposed boosting learning method is more effective in the error floor region than in the waterfall region. Therefore, its impact is more prominent with LDPC codes than with Polar or BCH codes, where the error floor doesn't manifest as prominently. However, boosting learning can be applied regardless of the code type and is also applicable targeting the waterfall region. To demonstrate this, we added simulation results of our method when applied to Polar and BCH in the waterfall region to the attached file's Table R.1. When compared to the SOTA model-based approach [15] in terms of the *waterfall performance*, a comparable performance is achieved. However, while [15] requires additional training/decoding complexities, our proposed method does not. In other words, our approach achieves “the *error floor SOTA* result” and exhibits comparable performance to “the *waterfall SOTA* result” [15] for other types of codes.
>
> ## Other types of noise
> The concept of training using uncorrected words can be consistently applied regardless of the channel type. In the attached file, Figure R.2, we included the results for the Rayleigh channel, showing the effectiveness of our proposed method holds across different channel types.
>
> ## Tie the weights
> The key mechanism for performance improvement is that the base and post decoders obtain decoding diversity by using different weights. Therefore, sharing weights between them would likely be ineffective. Exploring new weight sharing (or tying) techniques to reduce the model size can be an attractive avenue for future research.
>
> ## Multiple decoders
> It's possible to extend to the case of more than two decoders in the same manner. For instance, in the WiMax LDPC code result (Fig. 6(a)), uncorrected words from the error floor region of the base + post decoder (i.e., Eb/N0 5dB) can be collected to train a third decoder. In Figure R.1(a) of the attached file, we present numerical results that demonstrate the error floor's further reduction with the aid of this third decoder (labeled as “Base+Post+Third”). It's worth mentioning that introducing this third decoder might lead to increased decoding latency. Moreover, the process of collecting uncorrected words would be time-consuming, especially in the very low FER region. Addressing these challenges could be a promising direction for subsequent research.

---

> > ### Comment · Reviewer_zXhE · 2023-08-18
> >
> > Thank you addressing my concerns. The answers are satisfactory.

---

### Official Review · Reviewer_uaWt · 2023-07-11

**Soundness:** 2 fair
**Presentation:** 2 fair
**Contribution:** 2 fair
**Rating:** 5
**Confidence:** 4

**Summary:**


This paper presents novel training techniques for the NMS decoder of LDPC codes aimed at enhancing performance in the error floor region. The proposed decoding methods comprise two stages: a base decoder and a post decoder designed specifically for uncorrected codewords from the base decoder. In order to tackle the issue of vanishing gradient during training, a block-wise training schedule is introduced in this paper. By assigning distinct weights to unsatisfied check nodes, the error floor can be lowered with a small number of weights to be trained.

**Strengths:**

1. The impact of the proposed decoding technique is substantial, as evidenced by the figures which clearly demonstrate a significant improvement in code performance, particularly in the error floor region.
2. The analysis and interpretation of the results in Figure 3 were particularly valuable, providing a clear understanding of the distinctions between the methods employed for selecting training samples.

**Weaknesses:**

1. The proposed decoding method in this paper builds upon the concept of two-stage decoding and incorporates a post decoder that complements the base decoder. It is important to acknowledge that this idea is not entirely new within the field of coding theory for LDPC codes, as there has been prior research exploring similar approaches. The followings are some of the examples sharing a similar philosophy and it would be valuable to mention them as relevant previous work in this direction.
S. Yang, Y. Han, X. Wu, R. Wood and R. Galbraith, "A soft decodable concatenated LDPC code," 2015 IEEE International Magnetics Conference (INTERMAG), Beijing, China, 2015
J. Oh, J. Ha, H. Park and J. Moon, "RS-LDPC Concatenated Coding for the Modern Tape Storage Channel," in IEEE Transactions on Communications, vol. 64, no. 1, pp. 59-69, Jan. 2016
H. Park and J. Moon, "Improving SSD Read Latency via Coding," in IEEE Transactions on Computers, vol. 69, no. 12, pp. 1809-1822, 1 Dec. 2020

2. The clarity of the presentation style could be enhanced. For example, the specific problem addressed by the paper, such as the type of channel considered, is not explicitly stated. The reader must infer this information primarily from the related work section. Providing a clear and explicit description of the problem, including the type of channel under consideration, would greatly improve the overall clarity of the paper.

3. Nitpicks
- It would greatly enhance the presentation if the theoretical performance limit based on the code rate could be included in the plots, such as in Figures 3 and 6.
- Typo: On line 121, page 3, "protogrph" should be corrected to "protograph".

**Questions:**

In order to harness the potential of boosting learning, this study introduces a division of the decoding network into two components, with the post decoding stage aiding the base decoding process. A natural extension of this approach is to explore the possibility of incorporating more than two decoders. It would be intriguing to learn from the authors whether they have considered this scenario and if they can provide insights into the expected outcomes. Including discussions and potential answers to this question would enhance the manuscript's appeal and stimulate further interest in the research.

**Limitations:**

N/A - This work does not appear to have any discernible negative societal impact.

---

> ### Author Rebuttal · Authors · 2023-08-09
>
> Thank you for the valuable comments and the clarification.
>
> ## Related works sharing a similar philosophy
>
> Thank you for informing us about the related research. In those studies, they employ a two-stage decoding where the outer and inner decoders perform decoding subsequently or iteratively. Our proposed study bears similarities, as our study also carries out a two-stage decoding. However, unlike these concatenated codes, our method utilizes a single LDPC decoder. While the proposed decoder is conceptually divided into two stages (base/post) based on iterations, in practice, we employ only a single decoder with distinct weight parameter sets. Consequently, our approach avoids the need for the additional (or separate) parity bits that are typically associated with concatenated codes. Moreover, our approach has a novelty in that the post decoder is 'trained' dependent on the results of the base decoder. Nonetheless, we agree with the comment that our and previous works share the philosophy of complementary multiple decoders. We believe they are worth mentioning as relevant previous works.
>
>
> ## Clarity of the presentation
> As you mentioned, there was no explicit introduction of the channel model in the original manuscript. In the revised version, we will specify that the underlying channel is the AWGN channel. Additionally, we will add information about the sampling points of the post decoder and the size of the neural network.
>
> ## Performance limit
> As you suggested, we have added a graph showing the finite length capacity in the attached file, Figure R.1(a). The finite length performance limit was referenced from the paper [a].
>
> [a]: Y. Polyanskiy, ”Channel coding rate in the finite block-length regime,” IEEE Trans. on IT, vol. 56, no. 5, 2010
>
> ## Multiple decoders
> It's possible to extend to the case of more than two decoders in the same manner. For instance, in the WiMax LDPC code result (Fig. 6(a)), uncorrected words from the error floor region of the base + post decoder (i.e., Eb/N0 5dB) can be collected to train a third decoder. In Figure R.1(a) of the attached file, we present numerical results that demonstrate the error floor's further reduction with the aid of this third decoder (labeled as “Base+Post+Third”). It's worth mentioning that introducing this third decoder might lead to increased decoding latency. Moreover, the process of collecting uncorrected words would be time-consuming, especially in the very low FER region. Addressing these challenges could be a promising direction for subsequent research.

---

### Author Rebuttal · Authors · 2023-08-09

### Dear Reviewers and Area Chair,


We sincerely appreciate the time and effort you've taken to review our paper.

Your insightful feedback has undoubtedly enhanced our work.

We've carefully addressed each of your remarks and inquiries, providing detailed responses for each one.

We hope that our responses have addressed any concerns.

If you have further questions or need clarifications, please don't hesitate to ask.

---

> ### Comment · Reviewer_UbgL · 2023-08-13
>
> I thank the authors for the detailed response. I think the majority of my concerns have been addressed.

---

> ### Comment · Area_Chair_RPzh · 2023-08-18
> **Thank you for the rebuttal**
>
> Dear authors,
>
> thank you for providing a rebuttal. Some of the reviewers have already replied, so this is just to let you know that I am in contact with the remaining ones as well.
>
> Best,
> Your AC

---

### Decision · Program_Chairs · 2023-09-21

**Decision:**

Accept (poster)

**Comment:**

This paper studies how to optimise the training of neural-based min-sum decoders for LDPC codes. The objective is to reduce the error floor behaviour of the code (which is known to be an issue in applications targeting a very low block error rate).

The problem being tackled (i.e., reducing the error floor in LDPC decoding) is important, the impact of the proposed technique is substantial, and the numerical results are convincing. The rebuttal has helped to clarify some issues raised by the reviewers, and the final reviews are uniformly positive (although with different levels of enthusiasm). After my own reading of the reviews, rebuttal and paper, I agree with this positive evaluation and recommend acceptance. I encourage the authors to incorporate the comments and the additional simulation results in the camera ready.